# Promoting Healthcare Workers’ Adoption Intention of Artificial-Intelligence-Assisted Diagnosis and Treatment: The Chain Mediation of Social Influence and Human–Computer Trust

**DOI:** 10.3390/ijerph192013311

**Published:** 2022-10-15

**Authors:** Mengting Cheng, Xianmiao Li, Jicheng Xu

**Affiliations:** School of Economic and Management, Anhui University of Science and Technology, Huainan 232001, China

**Keywords:** performance expectancy, effort expectancy, social influence, human–computer trust, adoption intention, healthcare worker, AI-assisted diagnosis and treatment

## Abstract

Artificial intelligence (AI)-assisted diagnosis and treatment could expand the medical scenarios and augment work efficiency and accuracy. However, factors influencing healthcare workers’ adoption intention of AI-assisted diagnosis and treatment are not well-understood. This study conducted a cross-sectional study of 343 dental healthcare workers from tertiary hospitals and secondary hospitals in Anhui Province. The obtained data were analyzed using structural equation modeling. The results showed that performance expectancy and effort expectancy were both positively related to healthcare workers’ adoption intention of AI-assisted diagnosis and treatment. Social influence and human–computer trust, respectively, mediated the relationship between expectancy (performance expectancy and effort expectancy) and healthcare workers’ adoption intention of AI-assisted diagnosis and treatment. Furthermore, social influence and human–computer trust played a chain mediation role between expectancy and healthcare workers’ adoption intention of AI-assisted diagnosis and treatment. Our study provided novel insights into the path mechanism of healthcare workers’ adoption intention of AI-assisted diagnosis and treatment.

## 1. Introduction

Artificial intelligence (AI) is advertised as the principal general-purpose technology of this era [1,2]. Medical AI denotes applying a series of functions, including auxiliary diagnosis, risk prediction, disease triage, health management, and hospital management, through intelligent algorithms and technologies such as machine learning, representation learning, and deep learning [3]. Among them, AI-assisted diagnosis and treatment is highly followed worldwide, and domestic and foreign technology giants are striving in this field. AI robots perform medical AI-assisted diagnosis and treatment to complete daily supporting tasks, including directing methods, consultation in hospitals, image capture and recognition, assistive support in surgery, and epidemic-prevention information [4,5]. The last decade has witnessed a noticeable progression in the use of AI-assisted diagnosis and treatment in the field of dentistry [6].

AI-assisted diagnosis and treatment is the application of AI in disease diagnosis and treatment. For disease diagnosis, in oral implantology, several artificial intelligence models have been able to classify normal and osteoporotic subjects on panoramic radiographs, and their accuracy, sensitivity, and specificity are above 95%. They can assist doctors in identifying osteoporotic patients before implant treatment and improve the success rate of treatment [7]. In endodontics, some scholars [8,9] found that a deep-learning-based convolutional neural network algorithm can provide accurate diagnosis of dental caries, with an accuracy of 89.0% for premolars and 88.0% for molars, respectively, which is expected to be one of the effective methods for diagnosing caries. For treatment, dental robots play an important role in various fields of dentistry. In maxillofacial surgery, the robot can provide high-definition, three-dimensional magnified images and enter the body through a minimally invasive incision during surgery, which can significantly improve the precision, safety, and therapeutic effect of surgery [10]. In oral implantology, the robot system can achieve the reproduction of the anatomical structure of the surgical area, the precise design of preoperative implants, the automatic and precise implementation of surgery, and immediate implant restoration, which meet the requirements of precise, efficient, minimally invasive, and comfortable surgery [11]. Although the application of AI-assisted diagnosis and treatment could expand the medical scenarios [12,13] and augment work efficiency and accuracy [14,15], healthcare workers are unwilling to believe and rely on things that cannot be explained [16,17]. Owing to the lack of algorithm transparency, data security risks, uncertain medical responsibilities, and substitution threats, healthcare workers refuse to use AI [18,19]. Thus, it is imperative and urgent to investigate the acceptance of AI-assisted diagnosis and treatment for healthcare workers at this stage.

Research on technology-adoption intention in healthcare can be divided into three categories according to the different subjects of adoption: healthcare recipients (e.g., patients), healthcare workers (e.g., doctors, nurses), and healthcare institutions (e.g., hospitals, clinics). For different adopters, there are different factors influencing the intention to adopt technology, and the research models also differ [20,21,22,23,24,25]. This study compares the relevant literature, summarizes the theoretical basis and factors of healthcare workers’ intention to adopt technology, and lays the foundation for subsequent research on healthcare workers’ adoption intention of AI-assisted treatment technology (see Table 1).

The current research on healthcare workers’ adoption intention is primarily based on a single technology-adoption theory (e.g., technology-acceptance model (TAM) and unified theory of acceptance and use of technology (UTAUT)) [34,35], which explored the impact of AI technical characteristics [27], individual psychological cognition [26,28,29], and social norms [25,31,32] on healthcare workers’ intention to adopt AI technology. Of these, expectancy includes performance expectancy and effort expectancy, which are psychological cognitive factors that affect technology adoption [34,35,36,37]. Performance expectancy signifies the degree to which an individual believes that adopting new technology could improve his/her work performance and is an individual perception of the practicality of new technology [34,35]. Effort expectancy denotes the level of effort required by healthcare workers to use AI-assisted diagnosis and treatment and their perception of the ease of use of the new technology [34,35]. Two expectancies both positively affect users’ adoption intention [26,28,29]. Nevertheless, the adoption intention of AI by healthcare workers might change because of technological and environmental changes. At present, limited research has been conducted on the microprocess mechanism and medical scenarios of healthcare workers’ acceptance of AI.

It is worth noting that although existing studies have confirmed the validity of models such as TAM and UTAUT in assessing healthcare workers’ technology-adoption intention, medical AI is different from the previous technologies and presents the characteristics of high motility, high risk, and low trust. Therefore, a single model based on traditional TAM and UTAUT has a low explanation for intention to use AI [38]. Most existing studies have used mostly extended TAM or UTAUT to explore the factors influencing healthcare workers’ technology-adoption intention (see Table 1).

AI represents a highly capable, complex technology designed to mimic human intelligence, characterized by agency and control shifting from humans to technology and altering people’s previous understanding of the relationship between humans and technology, thereby creating a sense of trust [39]. Research has confirmed that HCT is an important prerequisite for user’s acceptance of medical AI, especially for more automated AI applications [40]. HCT contributes to reliability and the anthropomorphic features of AI [4,39,41]. Although many studies have examined trust in the interpersonal and societal domains, in different technologies, studies addressing trust in medical AI-assisted diagnosis and treatment are scarce. Madsen and Gregor (2000) defined HCT as “the extent to which a user is confident in, and willing to act on the basis of the recommendations, actions, and decisions of an artificially intelligent decision aid” [42], which enhances healthcare workers’ adoption intention of AI-assisted diagnosis and treatment [27,33]. Theories of interpersonal relationships have established trust as a social glue in relationships, groups, and societies [21,43]. However, the current literature leaves unanswered questions. For example, how is HCT built among healthcare workers, and how does it affect adoption intention? In addition, based on the UTAUT model, social influence exerts a positive impact on technology adoption [34,35]. The research has established that social influence indirectly affects users’ adoption intention of AI through trust [44]. Thus, based on the UTAUT model and HCT theory, this study investigates the path mechanism of expectancy on healthcare workers’ adoption intention of AI-assisted diagnosis and treatment.

This study contributes to the extant research literature in two ways. First, previous studies primarily used a single technology-adoption model to examine the electronic health record (EHR) [23,25,26] and telemedicine [24,29] by healthcare workers’ adoption intention. This study proposes an integrated model of the UTAUT model and HCT theory to determine what factors affect the intention of healthcare workers to adopt AI-assisted diagnosis and treatment, enriches the theoretical research of the UTAUT model, and expands the application scenarios of medical AI.

Second, previous research focused more on the direct impact of technology adoption [34,35]. The intermediary mechanism influencing the expectancy on healthcare workers’ adoption intention of AI-assisted diagnosis and treatment has received limited attention. Of note, HCT fails to elucidate the underlying mechanisms of why some healthcare workers are reluctant to believe medical AI. This study constructs a chain mediation model to illustrate the psychological mechanism of how healthcare workers’ expectancy affects their intention of embracing AI-assisted diagnosis and treatment. In addition, this study demonstrates the single mediating effect and chain mediation effect of social influence and HCT. By integrating social influence and HCT in the model, we offer a better understanding of how social influence and HCT can individually and collectively influence the association between expectancy and adoption intention for healthcare workers. Moreover, the conclusions could also provide a theoretical basis for medical explainable AI research and provide management enlightenment or reference for service providers, hospital managers, and government sectors.

The rest of the paper is structured as follows. Section 2 presents the proposed model with theoretical background. The research methodology is explained in Section 3. Data analysis and results are presented in Section 4. In Section 5, we discuss the implications of the findings, contributions, limitations, and directions for future research. Section 6 concludes the paper with some final thoughts.

## 2. Theoretical Background and Research Hypotheses

### 2.1. Theoretical Background

#### 2.1.1. The Unified Theory of Acceptance and Use of Technology

The UTAUT is a model to explain the generation of behavior proposed by integrating eight theoretical models, including the theory of reasoned action (TRA) [35,45], TAM [34], and theory of planned behavior (TPB) [46]. The UTAUT model could explain 70% of individual intentions to adopt information technology and 50% of information technology-adoption behavior. Among them, performance expectancy, effort expectancy, and social influence play a decisive role in individual intention, and facilitating conditions directly influence individual behavior [35]. This study focuses on healthcare workers’ adoption intention of AI-assisted diagnosis and treatment rather than on their adoption behavior. Therefore, the impact of facilitating conditions on healthcare workers’ adoption intention was not considered.

Wang et al. (2020) integrated UTAUT and task–technology fit (TTF) to understand the factors influencing consumer acceptance of healthcare wearable devices (HWDs). The key findings revealed that consumer acceptance is influenced by both users’ perceptions (performance expectancy, effort expectancy, social influence, and facilitating conditions) and the task–technology fit [20]. As the faster application of medical AI makes healthcare workers face greater uncertainty, the reasons for healthcare workers to adopt new technology are more diverse. Hossain et al. (2019) explored factors influencing the physicians’ adoption of EHR in Bangladesh and determined that social influence, facilitating conditions, and personal innovation positively influenced physicians’ intention to adopt the EHR system [25]. A Chinese study reported that doctors’ initial trust in AI-assisted diagnosis and treatment exerted a significant positive impact on doctors’ adoption intention [27]. The UTAUT model has been broadly used in healthcare.

#### 2.1.2. Human–Computer Trust Theory

The interaction between people and technology has special trust characteristics [47]. HCT is the degree to which people have confidence in AI systems and are willing to take action [42]. Trust is considered an attitude intention [47], which could directly influence acceptance and help people make cognitive judgments by decreasing risk perception [48] and enhancing benefit perception [49]. HCT is an attitude of trust that stems from the interaction between human and AI [50]. Fan et al. (2020) stated that perceived trust positively correlated with the adoption of AI-based medical diagnosis support system (AIMDSS) by healthcare professionals [27]. Furthermore, a Chinese study established that initial trust in an AI-assisted diagnosis system affects doctors’ adoption intention [37].

The traditional medical service relationship primarily occurs between patients and medical institutions or medical personnel, while in the medical AI scenario, the vital factor of human–technology interaction is added. HCT largely focuses on the collaboration between the human being and the automatic system [51]. From the perspective of technological object, the performance (such as trustworthiness and reliability) and attributes (such as appearance and sound) of the AI system itself as well as the different social and cultural situations might affect the HCT establishment [52]. From the viewpoint of technology users, people’s perceived expertise and responsiveness, risk cognition and brand perceptions, and other influencing factors constitute a preliminary model of technology trust, which emphasizes that users’ trust in AI chatbots could be a direct factor affecting users’ behavior [53].

### 2.2. Research Hypotheses

#### 2.2.1. Expectancy and Adoption Intention

Performance expectancy denotes the degree to which using technology would bring effectiveness to users in performing specific tasks [34,35]. In the context of AI-assisted diagnosis and treatment, performance expectancy indicates the extent to which AI-assisted diagnosis and treatment help healthcare workers increase their work efficiency. Effort expectancy is defined as the degree to which a person believes that using a particular system would be free of effort [34,35]. In this study, effort expectancy is to mirror healthcare workers’ perception of how easy it is to adopt AI-assisted diagnosis and treatment. Previous studies demonstrated that performance expectancy and effort expectancy are the primary determinants of intention to adopt a new technology [20,27,35]. Adenuga et al. (2017) posited that performance expectancy and effort expectancy exerted significant effects on Nigerian clinicians’ intention to adopt the telemedicine systems [29]. Regarding the adoption of the EHR [23,25] and the health information system (HIS) [28], studies have confirmed that performance expectancy and effort expectancy are positively related to physicians’ adoption intention. Hence, the following hypotheses are proposed:

**Hypothesis** **1a** **(H1a).**
*Performance expectancy is positively related to healthcare workers’ adoption intention of AI-assisted diagnosis and treatment.*


**Hypothesis** **1b** **(H1b).**
*Effort expectancy is positively related to healthcare workers’ adoption intention of AI-assisted diagnosis and treatment.*


#### 2.2.2. The Mediating Role of Social Influence

Social influence is defined as the degree to which an individual believes that important others believe he/she should adopt a new technology, which is considered the main predictor of general technology-acceptance behavior [34,35,54,55]. The rationale behind social influence could be that individuals want to fortify their relationships with critical persons by following their views of specific behaviors [56]. Based on the UTAUT model, Shiferaw and Mehari (2019) stated that social influence significantly and positively affected the intention of healthcare workers to use the electronic medical record system [57]. In addition, previous studies confirmed the positive association between social influence and behavioral intention [25,28]. Social influence can also aid the understanding of uncertainty reduction, as it might function as a substitute for interaction with the unknown and not-yet-available technology [58]. In other words, social influence is an active information-seeking method [59]. In this study, healthcare workers’ expectancy (performance expectancy and effort expectancy) of AI-assisted diagnosis and treatment are influenced by other healthcare workers’ attitudes, in turn influencing other healthcare workers’ attitudes toward AI-assisted diagnosis and treatment. This social interaction allows healthcare workers to gain information about AI-assisted diagnosis and treatment, reducing their perception of uncertainty and thus influencing their willingness to adopt AI-assisted diagnosis and treatment. Hence, the following hypotheses are proposed:

**Hypothesis** **2a** **(H2a).**
*Social influence mediates the relationship between performance expectancy and healthcare workers’ adoption intention of AI-assisted diagnosis and treatment.*


**Hypothesis** **2b** **(H2b).**
*Social influence mediates the relationship between effort expectancy and healthcare workers’ adoption intention of AI-assisted diagnosis and treatment.*


#### 2.2.3. The Mediating Role of Human–Computer Trust

Human–computer trust (HCT) denotes the beliefs that technology contributes to attaining personal goals and determining attitudes toward subsequent behavior in situations of uncertainty and vulnerability [60,61]. In the context of AI-assisted diagnosis and treatment, HCT demonstrates that healthcare workers believe the suggestions, actions, and decisions provided by AI-assisted diagnosis and treatment are reliable [62,63]. HCT can be viewed as a potential and critical prerequisite for the adoption of AI technology [40,64,65]. Nordheim et al. (2019) first developed an initial model of technology trust in a chatbot scenario and deduced that technology trust might be a direct factor influencing users’ adoption intention of AI chatbots [53]. Meanwhile, the association between expectancy and trust has been illustrated in the field of healthcare [27,66,67]. Furthermore, Prakash and Das (2021) surveyed 183 radiologists and demonstrated that trust played a mediating role between expectancy and adoption intention [68]. Of note, healthcare workers are more likely to trust AI-assisted diagnosis and treatment when they believe it would be more efficient or require less effort and then more likely to adopt it. Hence, the following hypotheses are proposed:

**Hypothesis** **3a** **(H3a).**
*HCT mediates the relationship between performance expectancy and healthcare workers’ adoption intention of AI-assisted diagnosis and treatment.*


**Hypothesis** **3b** **(H3b).**
*HCT mediates the relationship between effort expectancy and healthcare workers’ adoption intention of AI-assisted diagnosis and treatment.*


#### 2.2.4. The Chain Mediation Model of Social Influence and HCT

In the context of medical AI, the social reaction of medical AI would affect the trust and attitude of healthcare workers toward AI-assisted diagnosis and treatment. A study on trust in information systems also confirmed that social influence is related to HCT [44,54]. In addition, a study in China showed that social influence affects user behavior indirectly through trust [44,69]. Zhang et al. (2020) posited that in the automated vehicle sector, social influence manifests itself in the propaganda and assessment of users, which warrants service providers to attach importance to propaganda and word-of-mouth because the improvement of social acceptance could help enhance users’ trust and thus affect adoption intention [44]. Moreover, based on the above-mentioned discussion, users’ expectancy (performance expectancy and effort expectancy) influences others’ attitudes toward technology. When deciding whether to use AI-assisted diagnosis and treatment, healthcare workers will consider whether AI-assisted diagnosis and treatment could improve their work efficiency and whether the cost of learning AI-assisted diagnosis and treatment is less than the benefit [69,70]. Healthcare workers themselves are influenced by others’ attitudes toward AI-assisted diagnosis and treatment [71,72]. That is, when the person they think is critical to them has a positive attitude toward AI-assisted diagnosis and treatment, healthcare workers believe that AI-assisted diagnosis and treatment is reliable, accurate, and convenient [73,74]. When most people are negative about AI-assisted diagnosis and treatment, healthcare workers query the advice and decisions provided by AI-assisted diagnosis and treatment. Hence, the following hypotheses are proposed:

**Hypothesis** **4a** **(H4a).**
*The relationship between performance expectancy and healthcare workers’ adoption intention of AI-assisted diagnosis and treatment can be mediated sequentially by social influence and HCT.*


**Hypothesis** **4b** **(H4b).**
*The relationship between effort expectancy and healthcare workers’ adoption intention of AI-assisted diagnosis and treatment can be mediated sequentially by social influence and HCT.*


Figure 1 presents the theoretical model of this study.

## 3. Materials and Methods

### 3.1. Participants and Date Collection

Considering the service model of AI-assisted diagnosis and treatment, this study focused on the dental department, a medical scenario where AI was used widely, and selected healthcare professionals serving this department as the research subjects. Inclusion criteria consisted of dental healthcare workers who had a qualification certificate and a practice certificate, worked for at least 3 months, used AI-assisted diagnosis and treatment in the department, and understood the purpose of the survey, agreed, and participated voluntarily. A total of 450 questionnaires were distributed, of which 379 were collected, with a recovery rate of 84.2%. After screening the incomplete and visibly unqualified questionnaires, the total number of valid questionnaires was 343.

The demographic characteristics of participants were summarized in Table 2. Most participants were female (71.1%) with 77.0% aged < 41 years. Approximately half were married (47.5%), doctor (44.3%), and had a university degree (49.3%). The majority of the participants had less than 10 years of clinical experience (70.6%) and worked in tertiary hospital (57.7%).

Before the survey, participants were explicitly informed that the survey was for academic purposes only and that personal information (such as gender, age, and education) would be involved [75,76]. In the survey, participants were allowed to complete the questionnaire voluntarily and anonymously. After the survey, the survey data were kept in safe custody to protect the participants’ privacy.

### 3.2. Measures

All measurements were based on reliable mature scales. To ensure the applicability and efficacy of foreign scales in Chinese context, we strictly followed the “forward-backward translation” procedure [77]. Meanwhile, appropriate adjustments were made to the questions based on the AI-assisted diagnosis and treatment context, and two professionals were invited to examine the questionnaire for its clarity, terminology, logical consistency, and contextual relevance. The items and sources of the questionnaire are shown in Appendix A. All responses were reflected using a 5-point Likert scale, where 1 = strongly disagree, and 5 = strongly agree. Moreover, a pretest was administered to 20 dental healthcare workers before the formal research, and the questionnaire was revised to determine the official questionnaire based on the results of the research and feedback on the questions. To assess the reliability of our research instrument, Cronbach’s α values were calculated. Cronbach’s α of all the scales were greater than the threshold of 0.60 [78], which indicated that our research instrument had good reliability.

The 4-item performance expectancy scale was applied to measure performance expectancy [35]. A sample item from the questionnaire is “AI-assisted diagnosis and treatment will make my work more efficient”. The Cronbach’s α for this scale in the present study was 0.90. The 4-item effort expectancy scale was applied to measure effort expectancy [35]. A sample item from the questionnaire is “I can skillfully use AI-assisted diagnosis and treatment”. The Cronbach’s α for this scale in the present study was 0.93.

The 4-item social influence scale was applied to measure social influence [35]. A sample item from the questionnaire is “People who are important to me think that I should use AI-assisted diagnosis and treatment”. The Cronbach’s α for this scale in the present study was 0.92.

The 12-item human–computer trust scale developed by Gulati et al. (2018) was adapted [79]. A sample item from the questionnaire is “I can always rely on AI-assisted diagnosis and treatment”. The Cronbach’s α for this scale in the present study was 0.92.

The 3-item behavioral intention scale was applied to measure adoption intention [35]. A sample item from the questionnaire is “I intend to use AI-assisted diagnosis and treatment in the future”. The Cronbach’s α for this scale in the present study was 0.94.

### 3.3. Data Analysis

Before reliability and validity testing, this study implemented common methods bias (CMB) testing [80,81]. To minimize the threats of CMB, data confidentiality and anonymity, concealing variable names, and item mismatches were guaranteed, but it was still necessary to test the possible homologous variance. CMB was examined by Harman’s single-factor test [82,83]. Constraining the number of factors extracted to one, exploratory factor analysis yielded one single factor explaining 34.72% of the variance, lower than 50%, indicating there were no serious common method bias.

In this study, SPSS v22.0 was used to conduct the descriptive statistical analysis, internal reliability of the scales, and correlations between variables. AMOS 23.0 was used to conduct confirmatory factor analysis (CFA) and convergent validity of the scales. We used SPSS PROCESS macro 3.5 (MODEL 6) to test the chain mediation effect of social influence and HCT. The bootstrapping method produced 95% confidence intervals (CI) of these effects from 5000 bootstrap samples, which was the efficient method to test the mediating effect [84].

## 4. Results

### 4.1. Descriptive Statistics

Table 3 summarized the descriptive statistics and correlation among variables. Performance expectancy was significantly positively correlated with social influence (r = 0.583, *p* < 0.01), HCT (r = 0.559, *p* < 0.01), and adoption intention (r = 0.441, *p* < 0.01). Effort expectancy was significantly positively correlated with social influence (r = 0.391, *p* < 0.01), HCT (r = 0.558, *p* < 0.01), and adoption intention (r = 0.261, *p* < 0.01). Social influence had a positive correlation with HCT (r = 0.451, *p* < 0.01) and adoption intention (r = 0.551, *p* < 0.01). Similarly, there was a positive correlation between HCT and adoption intention (r = 0.604, *p* < 0.01).

### 4.2. Confirmatory Factor Analysis

We used CFA to show that the theoretical model had a good fit and establish the distinctiveness of study variables [85]. As shown in Table 4, the fitting degree of the one-factor model was poor, and our hypothesized five-factor model fits the data best (χ^2^/df = 2.213, GFI = 0.901, NFI = 0.937, RFI = 0.915, CFI = 0.964, RMSEA = 0.068). This indicated that the five-factor model had a good fit, and the distinctiveness of the five constructs in the current study was clear.

In addition, as shown in Table 3, the AVE of all constructs scored above the conventional value of 0.5, and the convergent validity of the model could be confirmed [86,87]. Discriminant validity was evaluated using the AVE square root calculated for every construct; all square roots were greater than the correlations among the constructs, proving discriminant validity (Table 3) [86].

### 4.3. Structural Model Testing

Figure 2 and Figure 3 indicated the results of the serial multiple mediation model. As shown in Figure 2, the total effect (β_b1_ = 0.634, *p* < 0.001) and the total direct effect (β_b2_ = 0.285, *p* < 0.001) of performance expectancy on healthcare workers’ adoption intention of AI-assisted diagnosis and treatment were found to be significant. Hence, H1a was supported. As shown in Figure 3, the total effect (β_d1_ = 0.594, *p* < 0.001) and the total direct effect (β_d2_ = 0.131, *p* < 0.05) of effort expectancy on healthcare workers’ adoption intention of AI-assisted diagnosis and treatment were found to be significant. Hence, H1b was supported.

Using the bootstrap method, we tested the mediating effects of social influence, HCT, and chain mediation effect of social influence and HCT, where the sampling value was set to 5000, and the CI was set to 95% [88]. Table 5 presents the results of the hypothesis testing.

For performance expectancy, the total indirect effect of performance expectancy and adoption intention was 0.349. The total indirect effect was significant at 95% CI (0.230, 0.481), excluding 0. Of these, indirect effect 1 was performance expectancy → social influence → adoption intention, which tested the mediating effect of social influence between performance expectancy and adoption intention. The effect value of indirect effect 1 was 0.261, with 95% CI (0.147, 0.386), excluding 0. Indirect effect 2 was performance expectancy → HCT → adoption intention, which tested the mediating effect of HCT between performance expectancy and adoption intention. The effect value of indirect effect 2 was 0.043, with 95% CI (0.010, 0.088), excluding 0. Indirect effect 3 was performance expectancy → social influence → HCT → adoption intention, which tested the chain mediation effect of social influence and HCT between performance expectancy and adoption intention. The effect value of indirect effect 3 was 0.045, with 95% CI (0.011, 0.090), excluding 0. Hence, H2a, H3a, and H4a were supported.

For effort expectancy, the total indirect effect of effort expectancy and adoption intention was 0.463. The total indirect effect was significant at 95% CI (0.332, 0.585), excluding 0. Of these, indirect effect 1 was effort expectancy → social influence → adoption intention, which tested the mediating effect of social influence between effort expectancy and adoption intention. The effect value of indirect effect 1 was 0.335, with 95% CI (0.213, 0.456), excluding 0. Indirect effect 2 was effort expectancy → HCT → adoption intention, which tested the mediating effect of HCT between performance expectancy and adoption intention. The effect value of indirect effect 2 was 0.088, with 95% CI (0.027, 0.157), excluding 0. Indirect effect 3 was effort expectancy → social influence → HCT → adoption intention, which tested the chain mediation effect of social influence and HCT between effort expectancy and adoption intention. The effect value of indirect effect 3 was 0.040, with 95% CI (0.012, 0.077), excluding 0. Hence, H2b, H3b, and H4b were supported.

## 5. Discussion

This study explored the adoption intention theoretical model of AI-assisted diagnosis and treatment by integrating the UTAUT model and HCT theory. The findings revealed that expectancy (performance expectancy and effort expectancy) positively influenced healthcare workers’ adoption intention of AI-assisted diagnosis and treatment, corroborating well-established evidence in previous UTAUT studies [20,23,25,27,28,29]. Notably, effort expectancy had a relatively smaller impact in determining healthcare workers’ adoption intention of AI-assisted diagnosis and treatment compared with performance expectancy. The reason might be that nowadays, the public has much experience in using high-tech devices. They might believe that they can handle AI-assisted diagnosis and treatment without spending too much effort. Moreover, if technology offers the needed functions, the public will accept more efforts in using it [89,90].

Our findings established that expectancy (performance expectancy and effort expectancy) influenced healthcare workers’ adoption intention through the mediation of social influence. The perceived utility and ease of use of AI-assisted diagnosis and treatment by healthcare workers would trigger positive attitudes among those around them [75,91]. When people engage in social interactions, healthcare workers are more likely to believe that adopting AI-assisted diagnosis and treatment is useful and effortless and thus would like to accept it. In addition, we found that expectancy (performance expectancy and effort expectancy) exerted a positive impact on adoption intention by the mediating effect of HCT, in line with previous studies [27,44,68,92]. That is, healthcare workers’ expectancy affected their trust in AI-assisted diagnosis and treatment, and subsequently, they would like to accept AI-assisted diagnosis and treatment.

This study supported the hypothesis that social influence and HCT played a chain mediation role between expectancy and healthcare workers’ adoption intention of AI-assisted diagnosis and treatment. Social influence was positively related to HCT, thereby supporting previous studies [27,44]. Fan et al. (2020) claimed that user advocacy and assessment are particularly crucial in the promotion and popularization of artificial intelligence-based medical diagnosis support system and that increased recognition of the service helps in enhancing users’ trust [27]. Healthcare workers could decrease costs of decision by referring to the attitudes of people around them toward AI-assisted diagnosis and treatment. Hence, healthcare workers’ expectancy was influenced by the positive attitudes of those around them toward technology. Furthermore, the positive impact would eventually transform into trust in AI-assisted diagnosis and treatment, resulting in the adoption intention of AI-assisted diagnosis and treatment by healthcare workers.

### 5.1. Theoretical Implications

The major implications of this study can be summarized as follows. First, this study enriches theoretical research on the application of medical AI scenarios. Previous research on healthcare workers’ intention to adopt technology focused on technologies such as the EHR [23,25,26], telemedicine [24,29], and the HIS [28]. However, limited research has been conducted on AI-assisted diagnosis and treatment. We extended scholarship by offering a theoretical framework and an empirically tested model of healthcare workers’ adoption intention of AI-assisted diagnosis and treatment, considering healthcare workers’ perception of AI-assisted diagnosis and treatment as well as the anticipated positive effects on work and society that AI may have. Perhaps this model will serve as a foundation for others seeking to understand the mixed attitudes and reactions of healthcare workers in the face of other medical AI scenarios [93].

Second, this study broadens the theoretical research of the UTAUT model by revealing the impact of human–computer trust (HCT) on healthcare workers’ adoption intention of AI-assisted diagnosis and treatment. Previous studies primarily used a single technology-adoption model, such as TAM and UTAUT, as the main research framework, establishing that performance expectancy and effort expectancy markedly affected the adoption intention of new technologies [34,35,94]. Nevertheless, medical AI differs from the previous technologies and presents the characteristics of high motility, high risk, and low trust. Our results suggested that HCT mediated the relationship between expectancy and healthcare workers’ adoption intention of AI-assisted diagnosis and treatment, similar to previous studies [27,44,68]. This study explains well how to promote HCT and sequentially accept AI-assisted diagnosis and treatment from the perspective of individual perception. This result is notable because it provides an explanation mechanism of HCT with regards to medical AI and extends the theoretical framework for UTAUT model in the medical field.

Third, this study addresses how performance expectancy and effort expectancy affected adoption intention of AI-assisted diagnosis and treatment, thereby enriching the research on the mediating mechanism between the above-mentioned relationships. While previous studies focused more on the direct effects of technology adoption, this study incorporated the HCT theory based on the UTAUT model and established a chain-mediated mechanism of social influence and HCT. Moreover, our findings tested the single mediating effect and chain mediating effect of social influence and HCT between expectancy and adoption intention, revealing interesting conversions between social influence and HCT. To some extent, the research conclusions also compensate for the low degree of explanation of AI-adoption intention based on single models, such as traditional TAM and UTAUT [38].

### 5.2. Practical Implications

The contributions of this study extend beyond the empirical findings and lie in the significance of its theoretical extension for the acceptance of AI-assisted diagnosis and treatment. First, service developers should focus on the performance expectancy and effort expectancy of healthcare workers for medical AI. Service developers should effectively comprehend the needs of healthcare workers for AI-assisted diagnosis and treatment, increase the R&D of AI-assisted diagnosis and treatment, develop professional functions that fulfill the needs of healthcare workers, ensure the accuracy of information and services provided by AI-assisted diagnosis and treatment, enhance work efficiency and service quality of healthcare workers, and improve the worthy perception of healthcare workers. Furthermore, service providers should be user-centered, focus on the experience of AI-assisted diagnosis and treatment, improve the simplicity of operation and interface friendliness, and enhance the perception of the ease of use for healthcare workers.

Second, hospital administrators could adjust their management strategies to augment the trust and acceptance of AI-assisted diagnosis and treatment among healthcare workers. For example, hospital managers should encourage healthcare workers to adopt the AI-assisted diagnosis and treatment as well as convey the hospitals’ support for the use of AI-assisted diagnosis and treatment. In addition, hospitals should conduct AI technology training for relevant healthcare workers to help them quickly understand AI-assisted diagnosis and treatment and enhance their expectancy [75]. Hospital managers could also associate work performance with salary and promotion for healthcare workers to urge them to adopt AI-assisted diagnosis and treatment.

Third, the government could amplify publicity on AI-assisted diagnosis and treatment and enhance its social influence. A study demonstrated that technology is adopted faster in mandatory settings [28]. The government’s vigorous promotion of AI-assisted diagnosis and treatment would enhance healthcare workers’ recognition of AI-assisted diagnosis and treatment. Moreover, social influence is a key factor affecting healthcare workers’ trust in AI-assisted diagnosis and treatment. When building trust with AI-assisted diagnosis and treatment, healthcare workers would refer to the positive or negative attitudes of those around them toward the technology. Thus, service providers must focus on their own publicity and word-of-mouth to increase the recognition of AI-assisted diagnosis and treatment, enhance HCT, and in turn influence healthcare workers’ adoption intention.

### 5.3. Limitations and Future Research

Although this study provided meaningful findings about healthcare workers’ adoption intention of AI-assisted diagnosis and treatment, the following points merit further research. First, our study used adoption intention instead of actual usage behaviors as the agent of acceptance because it is hard to measure potential users’ actual usage behavior in such a cross-sectional survey study, a fact that is commonly encountered by many previous studies [95]. A meta-analysis inferred that medium-to-large changes in intention induce small-to-medium changes in behavior [96]. Future studies could focus on healthcare workers’ adoption behavior of AI-assisted diagnosis and treatment to make the findings more practical. Second, this study was conducted in the dental department; thus, the findings might not be applicable to other departments. Further research might also involve other departments, such as imaging and clinic. Third, this study investigated the antecedents of healthcare workers’ adoption intention of AI-assisted diagnosis and treatment only from the viewpoint of healthcare workers’ perceptions. Notably, some other scenario-related factors (e.g., perceived risk and task–technology fit) might also contribute to healthcare workers’ acceptance and merit future explorations. Finally, this study was conducted with healthcare workers, including doctors, nurses, and medical technicians, but there were differences in the psychological perceptions of different types of healthcare workers (e.g., performance expectancy and effort expectancy). Research on the intention to adopt medical AI for just one type of healthcare worker would also be valuable in the future.

## 6. Conclusions

The adoption of AI-assisted diagnosis and treatment by healthcare workers could enhance work efficiency and accuracy; however, the “black box” nature of AI technology is a real barrier to its acceptance by healthcare workers. This study proposed and verified the theoretical model of adoption intention by integrating the UTAUT model and HCT theory to explain healthcare workers’ adoption intention of AI-assisted diagnosis and treatment. The results revealed that expectancy (performance expectancy and effort expectancy) positively affected healthcare workers’ adoption intention. In addition, we explored the single mediating effect and chain mediating effect of social influence and HCT between expectancy and adoption intention. This study could also effectively assist AI technology companies in their technology algorithm optimization, product development, and promotion and provide a reference for decision making on policy formulation for establishing trustworthy AI as well as the management needs of AI systems in government public and other sectors.

## Figures and Tables

**Figure 1 ijerph-19-13311-f001:**
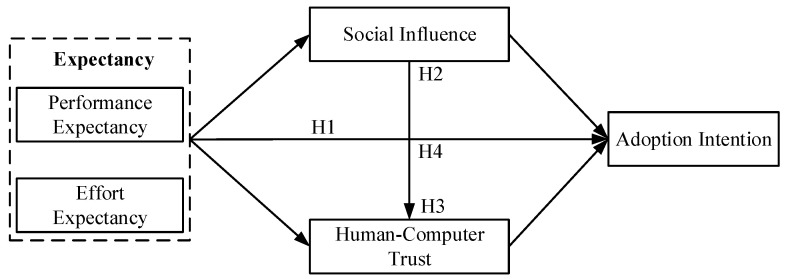
Research model.

**Figure 2 ijerph-19-13311-f002:**
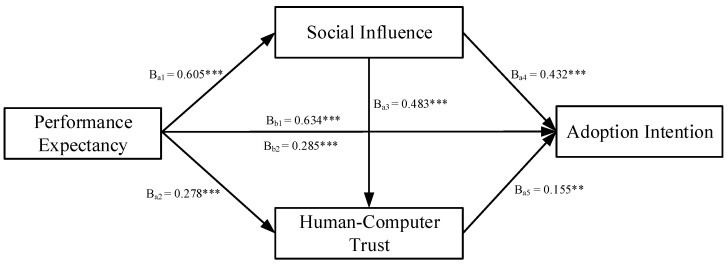
The serial multiple mediation model of performance expectancy. Note: a1, direct effect of performance expectancy on social influence; a2, direct effect of performance expectancy on human-computer trust; a3, direct effect of social influence on human-computer trust; a4, direct effect of social influence on adoption intention; a5, direct effect of human-computer trust on adoption intention; b1, total effect of performance expectancy on adoption intention; b2, direct effect of performance expectancy on adoption intention. ** *p* < 0.01; *** *p* < 0.001.

**Figure 3 ijerph-19-13311-f003:**
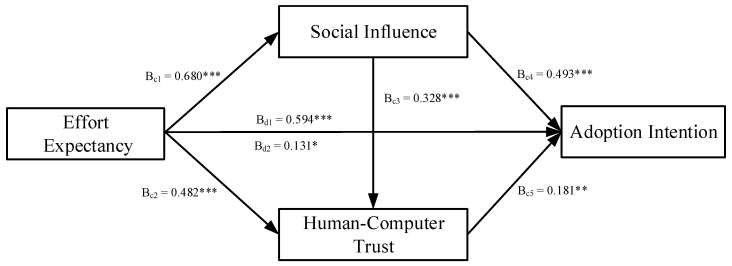
The serial multiple mediation model of effort expectancy. Note: c1, direct effect of effort expectancy on social influence; c2, direct effect of effort expectancy on human–computer trust; c3, direct effect of social influence on human–computer trust; c4, direct effect of social influence on adoption intention; c5, direct effect of human–computer trust on adoption intention; d1, total effect of effort expectancy on adoption intention; d2, direct effect of effort expectancy on adoption intention. * *p* < 0.05; ** *p* < 0.01; *** *p* < 0.001.

**Table 1 ijerph-19-13311-t001:** Literature review on healthcare workers’ adoption intention.

Authors	Context	Theoretical Basis	Region	Key Findings
Alsyouf et al. (2022) [23]	Nurses’ continuance intention of EHR	UTAUT, ECT, FFM	Jordan	Performance expectancy as a mediating variable on the relationships between the different personality dimensions and continuance intention, specifically conscientiousness as a moderator.
Pikkemaat et al. (2021) [24]	Physicians’ adoption intention of telemedicine	TPB	Sweden	Attitudes and perceived behavioral control being significant predictors for physicians to use telemedicine.
Hossain et al. (2019) [25]	Physicians’ adoption intention of EHR	Extended UTAUT	Bangladesh	Social influence, facilitating conditions, and personal innovativeness in information technology had a significant influence on physicians’ adoption intention to adopt the EHR system.
Alsyouf and Ishak (2018) [26]	Nurses’ continuance intention to use EHR	UTAUT and TMS	Jordan	Effort expectancy, performance expectancy, and facilitating conditions positively influence nurses’ continuance intention to use and top management support as significant and negatively related to nurses’ continuance adoption intention.
Fan et al. (2018) [27]	Healthcare workers’ adoption intention of AIMDSS	UTAUT, TTF, trust theory	China	Initial trust mediates the relationship between UTAUT factors and behavioral intentions.
Bawack and Kamdjoug (2018) [28]	Clinicians’ adoption intention of HIS	Extended UTAUT	Cameroon	Performance expectancy, effort expectancy, social influence, and facilitating conditions have a positive direct effect on clinicians’ adoption intention of HIS.
Adenuga et al. (2017) [29]	Clinicians’ adoption intention of telemedicine	UTAUT	Nigeria	Performance expectancy, effort expectancy, facilitating condition, and reinforcement factor have significant effects on clinicians’ adoption intention of telemedicine.
Liu and Cheng (2015) [30]	Physicians’ adoption intention of MEMR	The dual-factor model	Taiwan	Physicians’ intention to use MEMRs is significantly and directly related to perceived ease of use and perceived usefulness, but perceived threat has a negative influence on physicians’ adoption intention.
Hsieh (2015) [31]	Healthcare professionals’ adoption intention of health clouds	TPB and Status quo bias theory	Taiwan	Attitude, subjective norm, and perceived behavior control are shown to have positive and direct effects on healthcare professionals’ intention to use the health cloud.
Wu et al. (2011) [32]	Healthcare professionals’ adoption intention of mobile healthcare	TAM and TPB	Taiwan	Perceived usefulness, attitude, perceived behavioral control, and subjective norm have a positive effect on healthcare professionals’ adoption intention of mobile healthcare.
Egea and González (2011) [33]	Physicians’ acceptance of EHCR	Extended TAM	Southern Spain	Trust fully mediated the influences of perceived risk and information integrity perceptions on physicians’ acceptance of EHCR systems.

Note: EHR, electronic health record; AIMDSS, medical diagnosis support system; HIS, health information system; MEMR, mobile electronic medical records; EHCR, electronic health care records. UTAUT, unified theory of acceptance and use of technology; ECT, the theory of expectation confirmation; FFM, five-factor model; TPB, theory of planned behavior; TMS, top management support; TTF, task technology fit; TAM, technology-acceptance model.

**Table 2 ijerph-19-13311-t002:** Demographic characteristics for the participants (N = 343).

Characteristics	Frequency (f)	Percentage (%)
Gender		
Male	99	28.9
Female	244	71.1
Age		
≤20	37	10.8
21–30	105	30.6
31–40	122	35.6
41–50	57	16.6
≥51	22	6.4
Marital status		
Single	118	34.4
Married	163	47.5
Divorced	62	18.1
Education		
High school	27	7.9
Junior college	66	19.2
University	169	49.3
Master and above	81	23.6
Clinical experience		
<1	23	6.7
1–5	133	38.8
6–10	86	25.1
11–15	54	15.7
16–20	32	9.3
>20	15	4.4
Position		
Doctor	152	44.3
Nurse	124	36.2
Medical technician	67	19.5
Type of hospital		
tertiary	198	57.7
secondary	145	42.3

Note: In China, there are three hospital levels (the rank of hospitals). The best hospital level is level three (tertiary hospitals). Hospitals in this level can provide more beds, departments, professional nurses, professional doctors, and good service for patients. The higher the rank of hospital, the greater the use of AI.

**Table 3 ijerph-19-13311-t003:** Descriptive statistics and correlation among variables (N = 343).

	M	SD	AVE	PE	EE	SI	HCT	ADI
PE	3.96	0.75	0.697	0.835				
EE	3.11	0.97	0.779	0.276 **	0.883			
SI	3.53	0.75	0.800	0.583 **	0.391 **	0.894		
HCT	3.44	0.72	0.623	0.559 **	0.558 **	0.451 **	0.789	
ADI	3.70	0.72	0.650	0.441 **	0.261 **	0.511 **	0.604 **	0.806

Note: SD, standard deviations; AVE, average variance extracted; PE, performance expectancy; EE, effort expectancy; SI, social influence; HCT, human–computer trust; ADI, adoption intention. ** *p* < 0.05 (two-tailed). Values on the diagonal are the square root of the AVE of each variable.

**Table 4 ijerph-19-13311-t004:** Comparisons of measurement models.

Model	Variables	χ^2^/df	GFI	NFI	RFI	CFI	RMSEA
Five-factor model	PE, EE, SI, HCT, ADI	2.213	0.901	0.937	0.915	0.964	0.068
Four-factor model	PE + EE, SI, HCT, ADI	3.021	0.851	0.903	0.884	0.933	0.088
Three-factor model	PE + EE, SI + HCT, ADI	4.676	0.763	0.845	0.821	0.873	0.118
Two-factor model	PE + EE + SI + HCT, ADI	6.469	0.648	0.778	0.752	0.805	0.144
One-factor model	PE + EE + SI + HCT + ADI	10.870	0.481	0.621	0.583	0.642	0.194

Note: PE, performance expectancy; EE, effort expectancy; SI, social influence; HCT, human–computer trust; ADI, adoption intention. χ^2^/df, cmin/df; GFI, goodness-of-fit index; NFI, normed fit index; RFI, relative fit index; CFI, comparative fit index; RMSEA, root mean square error of approximation.

**Table 5 ijerph-19-13311-t005:** Direct and indirect effects.

Effect	X = PE			X = EE		
	Point Estimate	Boot SE	95%CI	Point Estimate	Boot SE	95%CI
Total indirect effect of X on ADI	0.349	0.064	[0.230, 0.481]	0.463	0.064	[0.332, 0.585]
Indirect 1: X → SI → ADI	0.261	0.061	[0.147, 0.386]	0.335	0.061	[0.213, 0.456]
Indirect 2: X → HCT → ADI	0.043	0.020	[0.010, 0.088]	0.088	0.033	[0.027, 0.157]
Indirect 3: X → SI → HCT → ADI	0.045	0.020	[0.011, 0.090]	0.040	0.017	[0.012, 0.077]

Note: CI, confidence interval; PE, performance expectancy; EE, effort expectancy; SI, social influence; HCT, human–computer trust; ADI, adoption intention.

## Data Availability

The datasets analyzed during the current study are not yet publicly available but are available from the corresponding authors upon reasonable request.

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
