# Peer review of "Promoting Healthcare Workers’ Adoption Intention of Artificial-Intelligence-Assisted Diagnosis and Treatment: The Chain Mediation of Social Influence and Human–Computer Trust"

_ijerph, 2022, doi:10.3390/ijerph192013311_

Round 1
Reviewer 1 Report
Title: Add the full name of acronym (AI). (Please be sure that all acronyms should be mentioned its full names when first mention).
Abstract: Add a statement addressing research methodology (data analysis).
Introduction: There is no explanation regarding the study objectives and novelty, the study questions should be outlined clearly, please consider it. Additionally, it is better to add a paragraph at the end of the introduction section which saying about the paper structure.
literature review: There is a missing part that authors need to add. The manuscript lacks a separate literature review section, which may provide a robust theoretical framework with regard to the research gap and hypotheses development as well as a description of existing studies on the topic. In this part, the authors link the study with prior conducted studies. That is, I propose to include a table showing how your study differs from others.
Theoretical Background: How did the authors select the two main factors for his selected study model (Performance expectancy and Effort expectancy)? and why they decided to skip the rest factors in UTAUT? and what was the reason behind integrating Human-Computer Trust Theory with UTAUT? Need more clarification on it.
Research Hypotheses:
1- Expectancy: How is the discussion about the hypothesis that does not have a significant effect? And how does it affect the overall research model built on this research?
2- The Mediating Role of Social Influence: Need more explanation and justification to address social influence as a mediating, what was your argument to consider this factor in particular as a mediator?. I suggest additional reading that will surely help increase the study’s general impact, taking into consideration the international perspective:
(Oldeweme, A.; Märtins, J.; Westmattelmann, D.; Schewe, G. The role of transparency, trust, and social influence on uncUnertainty reduction in times of pandemics: Empirical study on the adoption of COVID-19 tracing apps. J. Med. Internet Res. 2021, 23, e25893.)
(Alsyouf, A., Lutfi, A., Al-Bsheish, M., Jarrar, M. T., Al-Mugheed, K., Almaiah, M. A., ... & Ashour, A. (2022). Exposure Detection Applications Acceptance: The Case of COVID-19. International Journal of Environmental Research and Public Health, 19(12), 7307).
Materials and Methods: Materials and Methods are clearly described and the analysis is methodical and accurate. However, I think the detail of the indicators of convergent validity and discriminate validity should be listed.
Limitations and Future Research: It is necessary to add a clear and focused future research to be able to complete further research.
Finally, I suggest additional reading, mostly from MDPI journals, that will surely help increase the study’s general impact. References (to be consulted and not necessarily to be cited):
1-Alsyouf, A., & Ishak, A. K. (2018). Understanding EHRs continuance intention to use from the perspectives of UTAUT: Practice environment moderating effect and top management support as predictor variables. International Journal of Electronic Healthcare, 10(1-2), 24-59.
2- Jaber, M. M., Alameri, T., Ali, M. H., Alsyouf, A., Al-Bsheish, M., Aldhmadi, B. K., ... & Jarrar, M. T. (2022). Remotely monitoring COVID-19 patient health condition using metaheuristics convolute networks from IoT-based wearable device health data. Sensors, 22(3), 1205.
3- Siam, A. I., Almaiah, M. A., Al-Zahrani, A., Elazm, A. A., El Banby, G. M., El-Shafai, W., ... & El-Bahnasawy, N. A. (2021). Secure Health Monitoring Communication Systems Based on IoT and Cloud Computing for Medical Emergency Applications. Computational Intelligence and Neuroscience, 2021.
4- Alsyouf, A., Masa’deh, R. E., Albugami, M., Al-Bsheish, M., Lutfi, A., & Alsubahi, N. (2021). Risk of Fear and Anxiety in Utilising Health App Surveillance Due to COVID-19: Gender Differences Analysis. Risks, 9(10), 179.
Good luck
Reviewer 2 Report
Summary:
In the paper, titled "Promoting Healthcare Workers’ Adoption Intention of AI-assisted Diagnosis and Treatment: The Chain Mediation of Social Influence and Human-Computer Trust", the authors examined how healthcare professionals relate to technology that uses artificial intelligence to aid diagnosis and treatment. The authors describe the motivation behind the work and what theoretical model they used to investigate it. In particular, they point out that two existing theories, namely UTAUT (Unified Theory of Acceptance and Use of Technology) and HCT (Human-computer trust), are combined for the study. After setting up the model, it is shown how the investigation of the model was carried out, by means of a survey. The study design is described and the results are presented. Along the established hypotheses, the mediators of the established model are discussed and evaluated, limitation of the study are included. The authors conclude that the existing literature has been confirmed, and that no study of AI in this setting exists to date. They also summarize the factors that influence the acceptance of AI for diagnosis and treatment.
Points in favor of the paper:
- The paper is very well written.
- The paper draws a clear contribution
- The paper discusses related works
- The paper shows study results in a comprehensible manner
- The paper fits the scope of the journal
- The title of the paper fits the content
- The paper deals with a topical subject
- The paper results are sound und understandable
Points against the paper:
The paper shows interesting results and is also overall well crafted. However, there are a few points that are not clear to the reader
or should be improved:
- It is not apparent whether the participants have prior experience with AI?
-> In addition, the questionnaire should be shown somewhere, otherwise it is rather difficult to make an assessment.
- The group of test persons studied is heterogeneous in terms of use and expectations (a nurse certainly has different concerns than a doctor),
which needs to be discussed.
- In addition, it should be discussed how work from the area of XAI plays into this or has already shown findings that are related.
- From the title, diagnosis and treatment are very different especially related to AI, can you say something about this in the introduction;
was there a definition of AI for the participants of the study?
-> In this context, one should also again specifically address dentistry and how AI solutions look here
Round 2
Reviewer 1 Report
Thank you for addressing my comments and suggestions. Wishing you all the best.
Reviewer 2 Report
My concerns have been well addressed